# MonoidReduce: An Algebraic Framework for Memory-Efficient Neural Network Layers

## Abstract

Recent advances in memory-efficient neural network layers, such as FlashAttention, often appear as specialized engineering solutions but share a common mathematical structure. We show that many of these kernels can be understood as folds over commutative monoids, a perspective that unifies MapReduce-style computation with modern deep learning optimizations. Building on this, we introduce the Local Gradient Theorem, which provides a sufficient condition under which gradients of monoidal folds can be computed locally from the final output and individual inputs, enabling efficient backward passes. We demonstrate that attention, cross entropy, and two-layer MLPs all admit such monoid structures, recovering known memory-efficient kernels and extending the framework to new settings. This algebraic perspective offers a principled foundation for systematically designing memory- and cache-efficient layers, rather than discovering them in an ad-hoc manner.

## 1 Introduction

Training large neural networks is increasingly constrained by memory bandwidth and capacity rather than arithmetic throughput. In particular, attention layers and large-vocabulary classification heads require storing intermediate activations whose size grows quadratically or linearly with sequence length or vocabulary size. Recent work has shown that carefully reordering computation can eliminate the need to materialize these large intermediates. FlashAttention (Dao et al., 2022) introduced an IO-aware tiling strategy that computes exact attention in linear memory, later extended in FlashAttention-2 (Dao, 2023) for greater parallelism. In classification, Cut Cross-Entropy (CCE) (Wijmans et al., 2024) reorganizes the loss computation to avoid constructing full logit matrices. Liger kernels (Hsu et al., 2024) take a similar philosophy to operator fusion, providing efficient Triton implementations of common primitives. These works illustrate a pattern: memory efficiency can be achieved when a kernel can be expressed as a reduction.

A complementary line of work reduces memory through activation recomputation. General strategies such as gradient checkpointing (Chen et al., 2016; Griewank and Walther, 2000) selectively discard activations and recompute them during the backward pass, while reversible networks (Gomez et al., 2017) reconstruct hidden states exactly from later layers. IO-aware kernels such as FlashAttention (Dao et al., 2022; Dao, 2023) and Cut Cross-Entropy (Wijmans et al., 2024) can be viewed as specialized instances of the former: they deliberately recompute partial results in an online fashion to avoid materializing the full intermediate tensor in memory.

Our work generalizes this observation: we identify a broad class of *monoidal reductions* where such structured recomputation is always possible under a simple local gradient condition, and propose **MonoidReduce**, an abstraction that unifies FlashAttention, Cut Cross-Entropy, and related fused kernels, by observing that many of these algorithms can be written as *folds* over commutative monoids, enabling streaming and tiling. We prove a *Local Gradient Theorem* that characterizes when such folds admit recompute-friendly backwards passes. This provides a simple algebraic condition under which kernels can be made memory-efficient. A Proof of Concept implementation of this abstraction can be found in Appendix B.

Finally, our framework connects modern ML kernel design to classical systems and parallel algorithms. The fold view is closely related to MapReduce (Dean and Ghemawat, 2008) and prefix-sum

based reductions (Blelloch, 1990), providing a bridge between deep learning practice and foundational results in parallel computation. This is not novel, with recent work bridging this gap (Rush et al., 2024b;a). However, these works do not address the memory-efficient backwards passes that follow from the *Local Gradient Theorem*.

## 1.1 WHAT ARE COMMUTATIVE MONOIDS?

A monoid is a tuple $(T, \odot, \mathbf{id})$, where $T$ is a type, $\odot : T \to T \to T$ a binary operation on $T$, and $\mathbf{id}$ a term of type $T$. The binary operation must be associative, i.e. $a \odot (b \odot c) = (a \odot b) \odot c$, and $\mathbf{id}$ must be an identity element, i.e. $\mathbf{id} \odot a = a = a \odot \mathbf{id}$. Commutative monoids are monoids for which the binary operation is also commutative, that is, $a \odot b = b \odot a$.

There are many instances of commutative monoids in modern deep learning:

**Sum** Summation over real numbers:
$$T = \mathbb{R}$$
$$a \odot b = a + b$$
$$\mathbf{id} = 0$$

**WSum** Weighted sums of vectors.
$$T = \{v : \mathbb{R}^D, w : \mathbb{R}_{\geq 0}\}$$
$$a \odot b = c$$
$$\text{where } c_w = a_w + b_w$$
$$c_v = \begin{cases} a_v \frac{a_w}{c_w} + b_v \frac{b_w}{c_w}, & c_w > 0 \\ 0, & \text{otherwise} \end{cases}$$
$$\mathbf{id} = \{v : 0, w : 0\}$$

**LogWSum** Weighted sums of vectors with weights in logspace[1]:
$$T = \{v : \mathbb{R}^D, z : \mathbb{R}\}$$
$$a \odot b = c$$
$$\text{where } c_z = \ln(\exp(a_z) + \exp(b_z))$$
$$c_v = \begin{cases} a_v \exp(a_z - c_z) + b_v \exp(b_z - c_z), & c_z > -\infty \\ 0, & \text{otherwise} \end{cases}$$
$$\mathbf{id} = \{v : 0, z : -\infty\}$$

Given a commutative monoid $T$, we can derive a function $fold$ that takes as input a sequence of $T$s and returns the product over all elements in the sequence: $fold : [T] \to T = H \mapsto \bigodot_i H_i$.

## 2 MONOIDREDUCE

Commutative monoidal folds bring several advantages to large-scale machine learning. First, because the fold operation can be applied incrementally, there is no need to materialize all intermediate values in memory during the forward pass. This makes it possible to stream or distribute data through the computation, using GPU memory more efficiently, much like how MapReduce (Dean and Ghemawat, 2008) avoids keeping the entire dataset in memory by processing records in batches. Second, since partial results of a fold can be combined in any order, they can be computed independently across different devices (in the distributed setting) or thread blocks (in the kernel setting) and later merged, analogous to the "reduce" stage in MapReduce. Finally, if the derivative of the monoidal product ($\frac{d\ x \odot y}{d\ x}$) can be expressed in terms of $x \odot y$ and $x$, then the gradient of the entire fold inherits this property. This means gradients can be computed from the final aggregated value and *local* recomputation, enabling parallelization of the backward pass as well. This last observation is the main novelty presented in this paper, which we formalize as the *Local Gradient Theorem*,

---

[1]This monoid applies to attention: $\text{softmax}(A)V = x_0 \odot x_1 \odot \cdots$, where $x_i = \{v = V_i, z = A_i\}$

**Theorem 2.1** (Local Gradient Theorem). *Let $T$ be a finite-dimensional vector space and $(T, \odot, \mathbf{id})$ a commutative monoid with a derivative[2] $\frac{d\, x \odot y}{d\, x}$ that can be expressed as a function (D) of $x \odot y$ and $x$:*

$$\frac{d\, x \odot y}{d\, x} = D(x \odot y, x) \tag{1}$$

*We have that for any sequence $H : [T]$, the following holds:*

$$\frac{d\, fold(H)}{d\, H_i} = D(fold(H), H_i)$$

*Proof.* Let $P = fold(H) = \bigodot_i H_i$, for any partition of $H$ into $H^1, H^2$, with corresponding partial products $P_1 = fold(H^1) = \bigodot_i H_i^1$ and $P_2 = fold(H^2) = \bigodot_i H_i^2$.

$$P = P_1 \odot P_2 = P_2 \odot P_1 \qquad \text{(By commutativity and associativity)}$$
$$\frac{d\, P}{d\, P_1} = \frac{d\, P_1 \odot P_2}{d\, P_1}$$
$$= D(P_1 \odot P_2, P_1) \qquad \text{(By the condition from eq. 1)}$$
$$= D(P, P_1)$$

This extends trivially to $P_1 = H_i$, i.e., $H^1 = [H_i]$:

$$\frac{d\, P}{d\, H_i} = \frac{d\, P_1}{d\, H_i} \cdot D(P, P_1)$$

$$= D(P, P_1) \qquad\qquad \left(\frac{d\, P_1}{d\, H_i} = g \mapsto g\right)$$

$$= D(P, H_i)$$

$\square$

**Corollary 2.1.1.** *Consider the scenario where $H_{ij} = f(A_i, B_j)$ and let $P$ be the vector of final products ($P_i = fold(H_i) = \bigodot_j H_{ij}$). If we satisfy eq. 1, then we have that:*

$$\frac{d\, P}{d\, A_i} = \sum_j \frac{d\, H_{ij}}{d\, A_i} \cdot D(P_i, H_{ij}) \cdot \frac{d\, P}{d\, P_i}$$

$$\frac{d\, P}{d\, B_j} = \sum_i \frac{d\, H_{ij}}{d\, B_j} \cdot D(P_i, H_{ij}) \cdot \frac{d\, P}{d\, P_i}$$

*And more generally that for any partition of $A$ into $A^1, A^2, \ldots, A^m$ and $B$ into $B^1, B^2, \ldots, B^n$, with corresponding partitions of $H$ into $H^{11}, H^{12}, \ldots, H^{mn}$ and partial product vectors $P^{ij}$, where $P_k^{ij} = \bigodot_l H_{kl}^{ij}$, and let $P^i$ denote the vector of products $P_k^i = \bigodot_j P_k^{ij}$:*

$$\frac{d\, P}{d\, A^i} = \sum_j \frac{d\, P^{ij}}{d\, A^i} \cdot D(P^i, P^{ij}) \cdot \frac{d\, P}{d\, P^i}$$

$$\frac{d\, P}{d\, B^j} = \sum_i \frac{d\, P^{ij}}{d\, B^j} \cdot D(P^i, P^{ij}) \cdot \frac{d\, P}{d\, P^i}$$

---

[2]The mathematical convention used here is that $\frac{d\, A}{d\, B}$ denotes the linear map that transforms gradients of $A$ into gradients of $B$ (backward / pullback Jacobian), inspired by Conal Elliot's work on Automatic Differentiation (Elliott, 2009). Function composition $\cdot$ denotes sequential application of these linear maps, so the chain rule reads $\frac{d\, c}{d\, a} = \frac{d\, b}{d\, a} \cdot \frac{d\, c}{d\, b}$, corresponding to $(f \cdot g)(x) = f(g(x))$. $\langle a, b \rangle$ denotes dot product of vectors.

*Remark: $\frac{d\,P}{d\,P_i}$ and $\frac{d\,P}{d\,P^i}$ are simple diagonal projections.*

The implication of Corollary 2.1.1 is that gradients can be computed based only on local recomputation and final products, *not* the full $H$-matrix, which in turn facilitates highly parallel and cache-efficient kernels for backward passes.

In Appendix B we show a general proof of concept implementation of the MonoidReduce abstraction (listing 2), and an implementation of attention that uses this abstraction (listing 3), where forward and backward passes rely on user-supplied functions that map naturally to corollary 2.1.1. Table 1 summarizes these functions and their theoretical correspondence.

| User Function | Pass | Description |
|---|---|---|
| `proj_fold(A, B)` | Forward | Computes the partial fold vector $P^{ij}$ for slices $A^i$, $B^j$ of $A$ and $B$. |
| `proj_fold_bwd(A, B, P, gP)` | Backward | Computes local gradients $\frac{d\,P^{ij}}{d\,A^i} \cdot D(P^i, P^{ij})$ and $\frac{d\,P^{ij}}{d\,B^j} \cdot D(P^i, P^{ij})$. |
| `binary_reduce(P1, P2)` | Forward | The (vectorized) monoid operation $\odot$. |
| `init(A, B)` | Forward | Produces an appropriately shaped (monoidal) identity-valued tensor. |
| `islice(A)`, `jslice(B)` | Both | Produces partitions of the input, output, and gradients corresponding to $A^i$, $P^i$, $B^j$, etc. |

Table 1: User-supplied functions for batched MonoidReduce and their correspondence to corollary 2.1.1

**Forward pass.** Each slice pair $(A^i, B^j)$ is processed via `proj_fold` to produce partial results $P^{ij}$, which are then combined with (vectorized) `binary_reduce` to obtain the slice-level outputs $P^i$. This is embarrassingly parallel over $i$, but parallelizing over $j$ as well requires parallel reduction of the monoidal product $\odot$. A visualization of this can be seen in Figure 1.

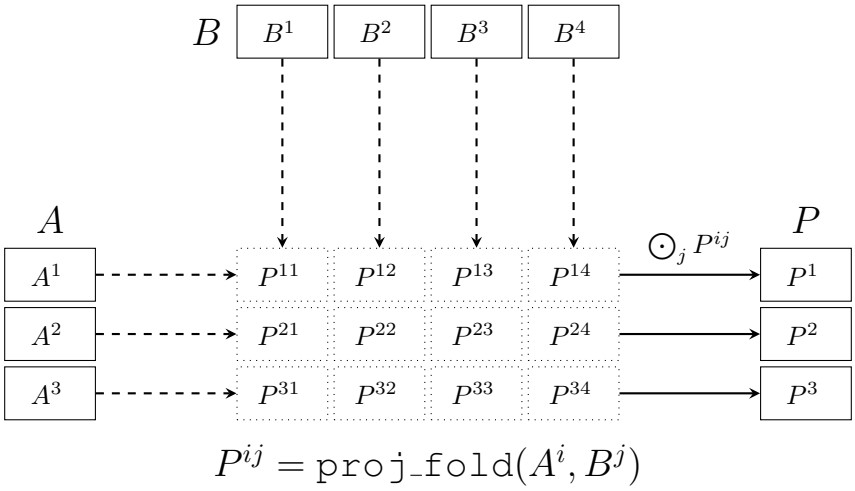

$$P^{ij} = \texttt{proj\_fold}(A^i, B^j)$$

Figure 1: A visualization of the MonoidReduce forward pass.

**Backward pass.** Gradients for each slice are computed using `proj_fold_bwd` for each pair of slices, and the results are added together across slices to form the full gradients for $A$ and $B$. The local gradient theorem (2.1) ensures that this is sound. Note that the summation of partial gradients poses a potential communication overhead. A visualization of this can be seen in Figure 2.

Corollary 2.1.1 also provides a general method for computing the Jacobian for $A$ during a fold over $B$, limited by how efficiently the Jacobian sum can be represented. In the case where $T$ is essentially a scalar (e.g. where $a \odot b = \text{logaddexp}(a, b)$ i.e. $fold(H) = \text{logsumexp}(H)$) this is manageable as the resulting Jacobian has the same shape as $A$, but for more complex monoids the

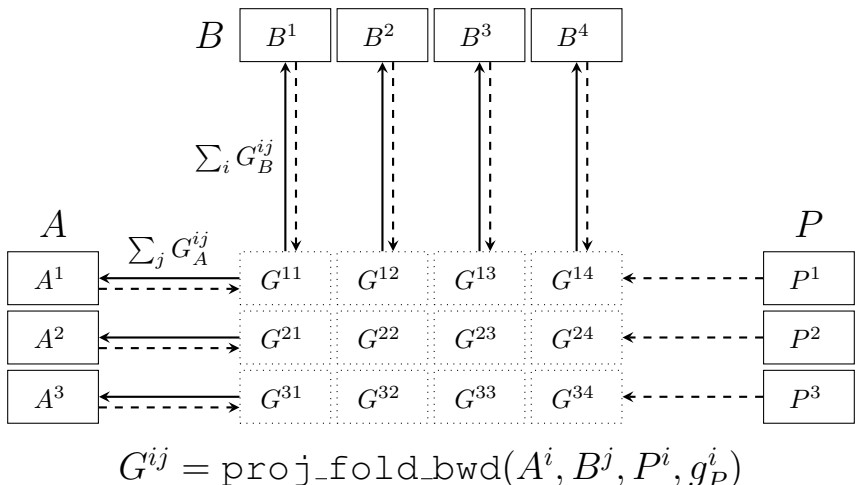

$$G^{ij} = \texttt{proj\_fold\_bwd}(A^i, B^j, P^i, g_P^i)$$

Figure 2: A visualization of the MonoidReduce backward pass.

cost of keeping the Jacobian in memory might be prohibitively expensive. When possible, such an approach simplifies the backward pass further, enabling us to apply the precomputed Jacobian to $\frac{d\mathcal{L}}{dP}$ to get the gradients for $A$, while separately computing the $B$-gradient by parallelizing over slices of $B$. As done in Hsu et al. (2024); Wijmans et al. (2024).

## 2.1 EXAMPLE: ATTENTION

The benefit of not materializing all partial values is most apparent in cases where the fold is applied to the results of matrix multiplications; one such instance is attention:

$$Q : \mathbb{R}^{M \times F}$$
$$K : \mathbb{R}^{N \times F}$$
$$V : \mathbb{R}^{N \times D}$$
$$\text{attention}(Q, K, V) = \text{softmax}(QK^\intercal)V$$
$$\text{attention}(Q, K, V)_i = \left( \bigodot_j f(A_i, B_j) \right)_v$$
$$\text{where}$$
$$A_i = Q_i$$
$$B_j = \{k : K_j, v : V_j\}$$
$$f(a, b) = \{z : \langle a, b_k \rangle, v : b_v\}$$
$$a \odot b = \begin{Bmatrix} z : & \ln(e^{a_z} + e^{b_z}) \\ v : & a_v e^{a_z - z} + b_v e^{b_z - z} \end{Bmatrix}$$

Here, we can think of $f$ as realizing an $M$ by $N$ matrix of $T$-values. Each $T$-value consists of a log-space weight $z$, and a value-vector $v$. The monoidal product $a \odot b$ results in a new $T$, with a weight corresponding to the log-space sum of $a$'s and $b$'s weight, and a value-vector corresponding to the weighted average of $a$'s and $b$'s value-vectors. Since the result is a fold over the $N$-dimension, the full matrix does not have to be materialized. We can compute appropriate chunks of the matrix, fold the chunk, and add to a running total, only realizing the $M$ normalizing factors, and the $M \times D$-sized weighted sum.

We also have that condition 1 holds:

$$c = a \odot b$$

$$\frac{d\,c}{d\,a} = g \mapsto \begin{pmatrix} z: & (g_z + \langle g_v, a_v - c_v \rangle) \exp(a_z - c_z) \\ v: & g_v \exp(a_z - c_z) \end{pmatrix}$$

$$= D(c, a)$$

Which due to Corollary 2.1.1 implies that the gradient w.r.t. $Q$, $K$ and $V$ can be calculated based on the final product $P$ and local (recomputed) $T$-values. Implementations of the necessary functions from Table 1 can be seen in Listing 1. Flash attention (Dao et al., 2022) can be seen as a highly optimized instance of this general approach.

Listing 1: Instances of functions from Table 1 for attention.

```
def init(a, b):
    aq, = a
    _, bv = b
    L, _ = aq.shape
    _, D = bv.shape
    z = aq.new_full((L,), float("-inf"))
    v = bv.new_zeros((L, D))

def proj_fold(a, b):
    aq, = a
    bk, bv = b
    logits = aq @ bk.t()
    z = torch.logsumexp(logits, dim=1)
    v = (logits - z[:, None]).exp() @ bv
    return z, v

def binary_reduce(x, y):
    xz, xv = x
    yz, yv = y
    z = torch.logaddexp(xz, yz)
    v = (xz - z).exp()[:, None] * xv + (yz - z).exp()[:, None] * yv
    return z, v

def D(global_p, local_p, gp):
    pz, pv = global_p
    lpz, lpv = local_p
    gz, gv = gp
    lgz = (gz + (gv * (lpv - pv)).sum(1)) * (lpz - pz).exp()
    lgv = gv * (lpz-pz).exp()[:, None]
    return lgz, lgv

def proj_fold_bwd(a, b, p, gp):
    # Fusing this would be more efficient.
    # The general form is used for presentation purposes.
    local_p, local_vjp = torch.func.vjp(proj_fold, a, b)
    return local_vjp(D(p, local_p, gp))
```

In Appendix C, Table 2, we give further examples of this approach applied to cross entropy against class indices[3], cross entropy between two distributions, and two-layer MLPs[4], all of which satisfy the condition for the local gradient theorem (Table 3).

## 3 COMPLEXITY ANALYSIS

The time and space complexity of MonoidReduce compared to a naive implementation depends on the time and space complexity of `proj_fold`, `proj_fold_bwd`, and `binary_reduce`, and the slicing of $A$ and $B$.

---

[3]Similar to both Liger Kernels and Cut Your Losses (Hsu et al., 2024; Wijmans et al., 2024)

[4]Similar to Tensor Parallelism in the multi-device setting (Shoeybi et al., 2019)

Consider inputs $A : \mathcal{A}^L$ and $B : \mathcal{B}^R$. We partition $A$ into $l$ equal-sized slices $A^1, \ldots, A^l$ and $B$ into $r$ equal-sized slices $B^1, \ldots, B^r$, let $|\mathcal{A}|$, let $|\mathcal{B}|$, and let $|\mathcal{P}|$ be the size of individual values of type $\mathcal{A}$, $\mathcal{B}$, and $\mathcal{P}$, respectively. Furthermore assume time and space complexities are (bi)linear with regards to $m$ and $n$: [5]

| | time | space | assumption |
|---|---|---|---|
| $\texttt{proj\_fold} : \mathcal{A}^m \times \mathcal{B}^n \to \mathcal{P}^m$ | $T_f(m, n)$ | $S_f(m, n)$ | bilinear |
| $\texttt{proj\_fold\_bwd} : \mathcal{A}^m \times \mathcal{B}^n \times \mathcal{P}^m \times \mathcal{P}^m \to \mathcal{A}^m \times \mathcal{B}^n$ | $T_b(m, n)$ | $S_b(m, n)$ | bilinear |
| $\texttt{binary\_reduce} : \mathcal{P}^m \times \mathcal{P}^m \to \mathcal{P}^m$ | $T_p(m)$ | $S_p(m)$ | linear |

### 3.1 FORWARD PASS

**Time Complexity.** The MonoidReduce forward pass uses $l \times r$ calls to $\texttt{proj\_fold}$, each taking time $T_f(\frac{L}{l}, \frac{R}{r})$, with $l \times (r - 1)$ reductions, each taking time $T_p(\frac{L}{l})$.

The total time complexity is therefore $l \times r \times T_f(\frac{L}{l}, \frac{R}{r}) + l \times (r - 1) \times T_p(\frac{L}{l})$, which by (bi)linearity simplifies to:

$$T_f(L, R) + (r - 1) \times T_p(L) \tag{2}$$

MonoidReduce thus incurs an extra cost of $(r - 1) \times T_p(L)$ compared to the full $\texttt{proj\_fold}$.

The critical path (with parallel reduction) takes time $\frac{T_f(L,R)}{lr} + \frac{T_p(L) \times \log r}{l}$, using Brent's Theorem (Brent, 1974) this translates to an $N$-worker parallel time complexity of:

$$\frac{T_f(L, R) + (r - 1) \times T_p(L)}{N} + \frac{T_f(L, R)}{lr} + \frac{T_p(L) \times \log r}{l} \tag{3}$$

**Space Complexity.** For any particular worker, we only need to realize $P^{ij}$, and aggregate it to the running total. Computing $P^{ij}$ uses $S_f(\frac{L}{l}, \frac{R}{r})$ space, and aggregation uses $S_p(\frac{L}{l})$ space. Assuming bilinearity, the per worker space complexity is therefore:

$$\frac{S_f(L, R)}{lr} + \frac{S_p(L)}{l} \tag{4}$$

A significant improvement compared to the full $\texttt{proj\_fold}$.

**Communication Complexity.** The computation of $P^{ij}$ requires reading $A^i$ and $B^j$, incurring a cost of $\frac{L}{l} \times |\mathcal{A}| + \frac{R}{r}|\mathcal{B}|$. The parallel sum $\bigodot_j P^{ij}$ incurs a cost of $(r - 1) \times \frac{L}{l} \times |P|$. Since we need to compute $l \times r$ $P^{ij}$-values, and $l$ parallel sums, this results in a total communication cost of:

$$r \times L \times |\mathcal{A}| + l \times R \times |\mathcal{B}| + (r - 1) \times L \times |\mathcal{P}| \tag{5}$$

MonoidReduce thus incurs an extra (total) communication cost of $(r-1) \times L \times (|\mathcal{A}| + |\mathcal{P}|) + (l-1) \times R \times |\mathcal{B}|$. This can be mitigated grouping tasks along the fold-axis, letting one worker be responsible for $W$ slices $B^j$, resulting in a lower communication cost of $\frac{r-W}{W} \times L \times (|\mathcal{A}| + |\mathcal{P}|) + (l-1) \times R \times |\mathcal{B}|)$ at the cost of worse idealized parallel time. This has been applied in, for example, FlashAttention (Dao, 2023).

### 3.2 BACKWARD PASS

**Time Complexity.** The analysis is similar to that of the forward pass, but instead of monoidal fold over $P^{ij}$, we aggregate gradients $\sum_i G_A^{ij}$ and $\sum_j G_B^{ij}$.

$$T_b(L, R) + L \times |\mathcal{A}| + R \times |\mathcal{B}| \tag{6}$$

---

[5]This is generally the case when $\texttt{proj\_fold}$ is dominated by matrix multiplications of $\mathbb{R}^{m \times k}, \mathbb{R}^{k \times n}$-shaped matrices, and it is the case for all examples in Table 2.

i.e., identical to the full `proj_fold_bwd` followed by accumulating the gradients.

The critical path (with parallel reduction) takes time $\frac{T_b(L,R)}{lr} + \frac{L}{l} \times |\mathcal{A}| \times \log r + \frac{R}{r} \times |\mathcal{B}| \times \log l$, using Brent's Theorem, this translates to an $N$-worker parallel time complexity of:

$$\frac{T_b(L,R) + L \times |\mathcal{A}| + R \times |\mathcal{B}|}{N} + \frac{T_b(L,R)}{lr} + \frac{L}{l} \times |\mathcal{A}| \times \log r + \frac{R}{r} \times |\mathcal{B}| \times \log l \quad (7)$$

**Space Complexity.** The analysis is similar to that of the forward pass. For any particular worker, we need only compute the local backward pass and add the resulting partial gradients to the global gradients:

$$\frac{S_b(L,R)}{lr} + \frac{L}{l}|\mathcal{A}| + \frac{R}{r}|\mathcal{B}| \quad (8)$$

A significant improvement compared to the full `proj_fold_bwd`.

**Communication Complexity.** The computation of $G^{ij}$ requires reading $A^i$, $B^j$, $P^i$, and $g_P^i$. Incurring a communication cost of $\frac{L}{l} \times |\mathcal{A}| + \frac{R}{r}|\mathcal{B}| + 2 \times \frac{L}{l}|\mathcal{P}|$.

The parallel sums incur a cost of $(r-1) \times \frac{L}{l} \times |\mathcal{A}|$ and $(l-1) \times \frac{R}{r} \times |\mathcal{B}|$.

This results in a total communication cost of:

$$2 \times r \times L \times (|\mathcal{A}| + |\mathcal{P}|) + 2 \times l \times R \times |\mathcal{B}| \quad (9)$$

The MonoidReduce backward pass thus incurs an extra communication cost of $2 \times (r-1) \times L \times (|\mathcal{A}| + |\mathcal{P}|) + 2 \times (l-1) \times R \times |\mathcal{B}|$. A way to mitigate this is to let one worker be responsible for several (input) slices $A^i$ or $B^j$, accumulating gradients along those slices locally. If a worker is responsible for $W$ $B^j$ (input) slices, thus accumulating $W$ $A$-gradients locally before adding to the global gradient, this reduces the communication overhead to $2 \times \frac{r-W}{W} \times L \times (|\mathcal{A}| + |\mathcal{P}|) + 2 \times (l-1) \times R \times |\mathcal{B}|$; again at the cost of worse idealized parallel time (critical path).

## 4 LIMITATIONS

While MonoidReduce provides an abstraction covering a wide range of existing memory-efficient implementations, and a *sufficient* condition for memory-efficient implementations in general, it does not apply universally. The central assumptions are that

- The reduction operator forms a commutative monoid.
- The derivative of the monoidal product can be expressed using only the combined value and one operand.
- The tradeoff between recomputation (or communication) cost and memory-efficiency is favorable.

This excludes many common operations, such as (sequences of) matrix multiplications, which can be understood as associative, *but not commutative*, folds. Instances where the tradeoff between recomputation (or communication) cost and memory-efficiency is not favorable are also common: Two-layer MLPs where the dimension being folded over is comparable in size to the output dimension will not see large memory-savings, and pay for the meager savings by incurring extra communication and computation.

From a systems perspective, recomputation introduces extra floating-point operations and potential hardware bottlenecks. In particular, as exemplified in Section 3, our generalized MonoidReduce example in listing 2 requires parallel fold of the monoidal product in the forward pass, and parallel sum of partial gradients in the backward pass; without careful tiling and shared-memory aggregation, atomic contention and communication costs can erode the benefits of theoretical memory savings. Furthermore, care must be taken with regard to numerical stability when implementing `proj_fold`, `proj_fold_bwd`, and `binary_reduce`.

Finally, our current work focuses on the algebraic formulation and theoretical guarantees rather than empirical benchmarking. Demonstrating practical wall-clock gains requires optimized implementations and careful IO analysis, as done in FlashAttention (Dao et al., 2022; Dao, 2023). As such, this paper should be considered mainly theoretical, highlighting the conceptual foundation for memory-efficient layers.

## 5 DISCUSSION

As previously stated, this paper is mainly theoretical. The working implementation given in Appendix B, while runnable and sometimes competitive against regular pytorch implementations on CPU, would require substantial work to go from Proof of Concept to fully fledged competitive framework. It is neither parallel nor very efficient. As hinted at in Section 3, slicing, grouping axis, and grouping size ($W$), all have significant impact on efficiency. A future framework would require automatic tuning of these parameters, as they are crucial for finding a performant tradeoff between increased communication and decreased memory pressure. However, given such a framework, it would enable faster iteration and innovation in memory and cache-efficient layers, designed by design, rather than by accident.

Also note that MonoidReduce can produce values that are used for aggregation but not passed onward in the computational graph, such as the normalizing factor $z$ in attention. This can enable more aggressive fusion of the `proj_fold_bwd` function and reduce communication cost. In a similar vein, if intermediate values used in the backward pass are shared within rows, they can be precomputed, decreasing both compute and communication costs. The framework as presented does not adequately address such optimizations.

## 6 FUTURE WORK

The most straightforward avenue for future work would be a working and efficient implementation of the proposed framework, e.g., a GPU-aware implementation of listing 2 integrated into pytorch, with support for automatic tuning. This would be a significant undertaking. While probably not matching the peak performance of highly optimized, hand-tuned, single-purpose kernels, a well-engineered framework could offer competitive performance for many layers and significantly accelerate the development of new memory-efficient architectures.

One such avenue is MLPs: Let $X, P, Q$ be matrices of shape $B \times D$, $K \times D$, and $K \times D$, respectively, where $B$ and $K$ are very large, but $D$ is small. Naive implementation would result in space complexity $BD + 2KD + BK$, with $BK$ being the dominant factor. MonoidReduce can remove the $BK$-term, at a cost of extra FLOPs during recomputation. The total FLOPs (forward and backward) for the naive implemntation is $12BKD$, whereas MonoidReduce increases this to $14BKD$. i.e. $\approx 17\%$ increase in FLOPs. If we let $K = B = 16384$, and $D = 128$, this would result in a memory requirement of $\approx 2\%$ of the naive implementation, making MLPs for large batches ($B$) and large hidden dimension ($K$), but small input and output dimension ($D$), much more feasible.

## 7 LLM USAGE

LLMs have been used in this work to flesh out drafts of the Introduction, Limitations, and Abstract, which were later edited. LLMs have also been used for general feedback and criticism.

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

## A  NOTATION

**Types & Terms**  $a : A$ denotes that the term $a$ has type $A$. For example, $a : \mathbb{R}$ denotes that the term $a$ is a real number.

**Functions**  $A \to B$ denotes the type of functions from $A$ to $B$. Lambda functions are written out as $f = x \mapsto x^2 + 3$, meaning $f(x) = x^2 + 3$.

$\to$ is interpreted in a right-associative manner, meaning $A \to B \to C = A \to (B \to C)$, which is isomorphic to $A \times B \to C$.

**Product Types**  $A \times B$ denotes the product type of $A$ and $B$. For brevity, terms of product types are occasionally written out as tuples. If $a : A$ and $b : B$, we can write $(a, b) : A \times B$, and vice versa. For a term $x : A \times B$, $x_0$ is the term corresponding to the left-hand type ($A$), and $x_1$ corresponds to the right-hand type. We also used named product types, where $\{a : A, b : B\}$ is the type of pairs of $A$ and $B$ indexed by the names $a$ and $b$, i.e. $X : \{a : A, b : B\}$ says that $X$ is a named product type, with two named parts $a$ and $b$ of type $A$ and $B$ respectively. Similarly, terms of named product types are written out as $x = \{a : v, b : w\}$, meaning $x_a = v$, $x_b = w$.

## B  BATCHED MONOIDREDUCE

Listing 2: A working, generalized, batched, Proof of Concept of MonoidReduce, implemented as a torch.autograd.Function-factory.

```python
import torch
from torch.autograd import Function
from torch.autograd.function import once_differentiable

def MonoidReduce(
    fun_name,
    *,
    init, # construct identity-tensors based on a and b
    avals, # number of A-tensors
    bvals, # number of B-tensors
    islice, # slicing over i-axis
    jslice, # slicing over j-axis
    proj_fold, # fused projection and fold over A and B-slices.
    proj_fold_bwd, # fused projection and fold backwards op.
    binary_reduce, # monoidal product
    ):
    """
    Constructor for MonoidReducers.
    """

    #aslice = slice(0, avals)
    astop = avals
    bstop = astop + bvals

    class DynamicFunction(torch.autograd.Function):
        @staticmethod
        def forward(*inputs):
            a = inputs[0:astop]
            b = inputs[astop:bstop]
            p = init(a, b)
            # Running over ai can be done in parallel
            # (requires rework of how to construct final p)
            for ai, pi in zip(
                    islice(a),
                    islice(p),
                    ):
                pi_acc = pi
                # Each bj slice can be run in parallel, requires
                # parallel reduction of the pij-values over j.
                for bj in jslice(b):
                    pij = proj_fold(ai, bj)
                    pi_acc = binary_reduce(pi_acc, pij)

                for pi_p, pi_acc_p in zip(pi, pi_acc):
                    pi_p.copy_(pi_acc_p)

            return p

        @staticmethod
        def setup_context(ctx, inputs, outputs):
            ctx.save_for_backward(
                *inputs, *outputs
            )

        @staticmethod
        @once_differentiable
        def backward(ctx, *gp):
            saved = ctx.saved_tensors

            a = saved[:astop]
            b = saved[astop:bstop]
```

```
                    p = saved[bstop:]

                    ga = [a_p.new_zeros(a_p.shape) for a_p in a]
                    amask = [a_p.requires_grad for a_p in a]
                    gb = [b_p.new_zeros(b_p.shape) for b_p in b]
                    bmask = [b_p.requires_grad for b_p in b]
                    # Running over ai can be done in parallel
                    # (requires atomic adds or other parallel sum over gaij)
                    for ai, gai, pi, gpi in zip(
                            islice(a),
                            islice(ga),
                            islice(p),
                            islice(gp),
                            ):
                        # Running over bi can be done in parallel
                        # (requires atomic adds or other parallel sum over gbij)
                        for bj, gbj in zip(
                                jslice(b),
                                jslice(gb),
                                ):
                            lgaij, lgbij = proj_fold_bwd(ai, bj, pi, gpi)
                            for gai_p, lgaij_p, requires_grad in zip(gai, lgaij, amask):
                                if requires_grad:
                                    gai_p.add_(lgaij_p)
                            for gbj_p, lgbij_p, requires_grad in zip(gbj, lgbij, bmask):
                                if requires_grad:
                                    gbj_p.add_(lgbij_p)

                return *ga, *gb

        DynamicFunction.__name__ = f'MonoidReduce_{fun_name}'
        DynamicFunction.__qualname__ = f'MonoidReduce_{fun_name}'
        DynamicFunction.__module__ = getattr(init, '__module__', __name__)

        return DynamicFunction
```

Listing 3: An implementation of attention using the MonoidReduce-factory from listing 2.

```
def init(a, b):
    aq, = a
    _, bv = b
    B = aq.shape[0]
    D = bv.shape[1]

    z = aq.new_full((B,), float('-inf'))
    v = bv.new_zeros((B, D))
    return z, v

def islice(parts):
    return zip(*(part.chunk(8) for part in parts))

def proj_fold(a, b):
    aq, = a
    bk, bv = b
    logits = aq @ bk.t()
    z = torch.logsumexp(logits, dim=1)
    v = (logits - z[:, None]).exp() @ bv
    return z, v

def binary_reduce(x, y):
    xz, xv = x
    yz, yv = y
    z = torch.logaddexp(xz, yz)
    v = (xz - z).exp()[:, None] * xv + (yz - z).exp()[:, None] * yv
    return z, v
```

```python
def proj_fold_bwd(a, b, p, gp):
    # ignore gz, as it is only used for aggregation.
    aq, bk, bv, pz, pv, _, gv = *a, *b, *p, *gp
    # Recompute relevant stuff
    logits = aq @ bk.t()
    ws = (logits - pz[:, None]).exp()
    # calculate gradients.
    # (gv * pv).sum(1) could be precomputed before broadcast,
    gbv = ws.t() @ gv
    glogits = (gv @ bv.t() - (gv * pv).sum(1)[:, None]) * ws
    gaq = glogits @ bk
    gbk = glogits.t() @ aq

    return (gaq,), (gbk, gbv)

Attention = MonoidReduce('Attention',
                         init=init,
                         avals=1,
                         bvals=2,
                         islice=islice,
                         jslice=islice,
                         proj_fold=proj_fold,
                         proj_fold_bwd=proj_fold_bwd,
                         binary_reduce=binary_reduce
                         )

def attention(q, k, v):
    """
    q: L x DI
    k: R x DI
    v: R x DO
    """
    z, v = Attention.apply(q, k, v)
    return v
```

Listing 4: An implementation of cross-entropy between two parameterized distributions using the MonoidReduce-factory from listing 2.

```python
def init(a, b):
    ap, _ = a
    B = ap.shape[0]
    p = ap.new_full((B,), float('-inf'))
    q = ap.new_full((B,), float('-inf'))
    n = ap.new_zeros((B,))
    return p, q, n

def islice(parts):
    return zip(*(part.chunk(8) for part in parts))

def proj_fold(a, b):
    ap, aq = a
    bc, bd = b
    pl = ap @ bc.t()
    ql = aq @ bd.t()

    p = torch.logsumexp(pl, dim=1)
    q = torch.logsumexp(ql, dim=1)
    n = ((ql - q[:, None]).exp() * pl).sum(1)
    return p, q, n

def binary_reduce(x, y):
    xp, xq, xn = x
    yp, yq, yn = y
```

```
        p = torch.logaddexp(xp, yp)
        q = torch.logaddexp(xq, yq)
        n = xn * (xq - q).exp() + yn * (yq - q).exp()
        return p, q, n

def proj_fold_bwd(a, b, p, gp):
    (lp, lq, ln), local_vjp = vjp(proj_fold, a, b)

    p, q, n = p
    gp, gq, gn = gp
    # Compute d p^{ij} / d [a, b].

    lgp = gp * (lp - p).exp()
    lgq = (gq + gn * (ln - n)) * (lq - q).exp()
    lgn = gn * (lq - q).exp()
    lg = (lgp, lgq, lgn)

    # Run local vjp.
    return local_vjp(lg)

XEntropy2 = MonoidReduce(
    'XEntropy2',
    init=init,
    avals=2,
    bvals=2,
    islice=islice,
    jslice=islice,
    proj_fold=proj_fold,
    proj_fold_bwd=proj_fold_bwd,
    binary_reduce=binary_reduce)

def xentropy2(p, c, q, d):
    p, _, n = XEntropy2.apply(p, q, c, d)
    return p - n
```

The code presented in Listing 2, while functional, should be regarded as a Proof Of Concept. It is neither parallel nor very efficient. It does, however, showcase the memory-efficiency aspects: Figures 3 and 4 show memory profiles of attention (Listing 3) using the above implementation (extracted using `torch.profile` with cpu-pytorch).

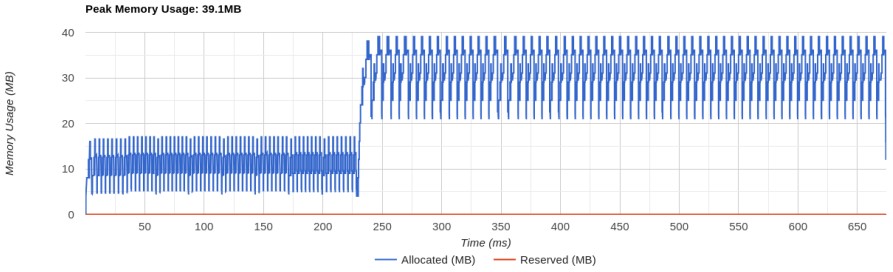

Figure 3: Memory profile for MonoidReduce attention (forward + backward).

## C   MONOIDS CORRESPONDING TO CROSSENTROPY AND TWO-LAYER MLPS

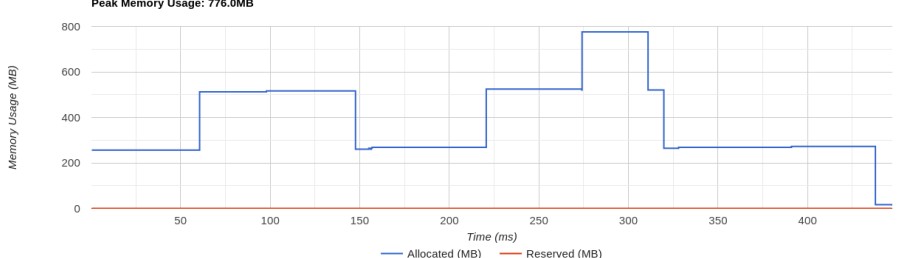

Figure 4: Memory profile for naive attention: `softmax(q @ k.t())@v`, (forward + backward)

| Input | Map | Reduce |
|---|---|---|

**Attention:** $Y = \text{softmax}(QK^\intercal)V$

| Input | Map | Reduce |
|---|---|---|
| $Q : \mathbb{R}^{M \times F}$ | $T : \{z : \mathbb{R}, v : \mathbb{R}^D\}$ | $P : T^M$ |
| $K : \mathbb{R}^{N \times F}$ | $A_i = Q_i$ | $P_i = \bigodot_j f(A_i, B_j)$ |
| $V : \mathbb{R}^{N \times D}$ | $B_j = \{k : K_j, v : V_j\}$ | |
| $\downarrow$ | $f(a,b) = \begin{cases} z : & \langle a, b_q \rangle \\ v : & b_v \end{cases}$ | $a \odot b = \begin{cases} z : & \ln(e^{a_z} + e^{b_z}) \\ v : & a_v e^{a_z - z} + b_v e^{b_z - z} \end{cases}$ |
| $Y : \mathbb{R}^{M \times D}$ | | $Y_i = P_{iv}$ |

**Cross Entropy:** $Y = \text{cross-entropy}(\text{logits} = PC^\intercal, \text{targets} = T)$

| Input | Map | Reduce |
|---|---|---|
| $P : \mathbb{R}^{M \times D}$ | $T : \{p : \mathbb{R}, n : \mathbb{R}\}$ | $P : T^M$ |
| $C : \mathbb{R}^{N \times D}$ | $A_i = \{p : P_i, t : T_i\}$ | $P_i = \bigodot_j f(A_i, B_j)$ |
| $T : N^M$ | $B_j = \{c : C_j, ix : j\}$ | |
| $\downarrow$ | $f(a,b) = \begin{cases} p : & \langle a_p, b_c \rangle \\ n : & a_t = b_{ix} \; ? \; p \; : \; 0 \end{cases}$ | $a \odot b = \begin{cases} p : & \ln(e^{a_p} + e^{b_p}) \\ n : & a_n + b_n \end{cases}$ |
| $Y : \mathbb{R}^M$ | | $Y_i = P_{ip} - P_{in}$ |

**Cross Entropy:** $Y = \text{cross-entropy}(\text{logits} = PC^\intercal, \text{targets} = \text{softmax}(QD^\intercal))$

| Input | Map | Reduce |
|---|---|---|
| $P : \mathbb{R}^{M \times D}$ | $T : \{p : \mathbb{R}, q : \mathbb{R}, n : \mathbb{R}\}$ | $P : T^M$ |
| $C : \mathbb{R}^{N \times D}$ | $A_i = \{p : P_i, q : Q_i\}$ | $P_i = \bigodot_j f(A_i, B_j)$ |
| $Q : \mathbb{R}^{M \times E}$ | $B_j = \{p : C_j, q : D_j\}$ | |
| $D : \mathbb{R}^{N \times E}$ | $f(a,b) = \begin{cases} p : & \langle a_p, b_p \rangle \\ q : & \langle a_q, b_q \rangle \\ n : & p \end{cases}$ | $a \odot b = \begin{cases} p : & \ln(e^{a_p} + e^{b_p}) \\ q : & \ln(e^{a_q} + e^{b_q}) \\ n : & a_n e^{a_q - q} + b_n e^{b_q - q} \end{cases}$ |
| $\downarrow$ | | |
| $Y : \mathbb{R}^M$ | | $Y_i = P_{ip} - P_{in}$ |

**MLP:** $Y = \sigma(XP^\intercal)Q$

| Input | Map | Reduce |
|---|---|---|
| $X : \mathbb{R}^{B \times M}$ | $T : \mathbb{R}^N$ | $P : T^B$ |
| $P : \mathbb{R}^{K \times M}$ | $A_i = X_i$ | $P_i = \bigodot_j f(A_i, B_j)$ |
| $Q : \mathbb{R}^{K \times N}$ | $B_j = \{p : P_j, q : Q_j\}$ | |
| $\downarrow$ | $f(a,b) = \sigma(\langle a, b_p \rangle) b_q$ | $a \odot b = a + b$ |
| $Y : \mathbb{R}^{B \times N}$ | | $Y_i = P_i$ |

Table 2: Examples of Monoids $T$, map functions from inputs to intermediate values, reduction operations, and out projections that realize different operations used in Artificial Neural Networks.

| Monoid | Gradient ($D$) |
|---|---|

### CrossEntropy

$$T : \{p : \mathbb{R}, n : \mathbb{R}\}$$

$$a \odot b = \begin{Bmatrix} p : & \ln(e^{a_p} + e^{b_p}) \\ n : & a_n + b_n \end{Bmatrix}$$

$$= c$$

$$\frac{d\,c}{d\,a} = g \mapsto \begin{Bmatrix} p : & g_p e^{a_p - c_p} \\ n : & g_n \end{Bmatrix}$$

### CrossEntropy between two distributions

$$T : \{q : \mathbb{R}, p : \mathbb{R}, n : \mathbb{R}\}$$

$$a \odot b = \begin{Bmatrix} p : & \ln(e^{a_p} + e^{b_p}) \\ q : & \ln(e^{a_q} + e^{b_q}) \\ n : & a_n e^{a_q - q} + b_n e^{b_q - q} \end{Bmatrix}$$

$$= c$$

$$\frac{d\,c}{d\,a} = g \mapsto \begin{Bmatrix} p : & g_p e^{a_p - c_p} \\ q : & (g_q + g_n(a_n - c_n))e^{a_q - c_q} \\ n : & g_n e^{a_q - c_q} \end{Bmatrix}$$

### MLP

$$T : \{v : \mathbb{R}^D\}$$

$$a \odot b = \{v : a_v + b_v\}$$

$$= c$$

$$\frac{d\,c}{d\,a} = g \mapsto g$$

Table 3: Proofs that condition 1 holds for the given monoids, and by extension that Theorem 2.1 and all its corollaries hold.