# OpenReview forum: "MonoidReduce: An Algebraic Framework for Memory-Efficient Neural Network Layers"
_ICLR.cc/2026/Conference — Submitted to ICLR 2026_

### Official Review · Reviewer_UHrH · 2025-10-22

**Soundness:** 2
**Presentation:** 1
**Contribution:** 1
**Rating:** 2
**Confidence:** 5

**Summary:**

This paper introduces MonoidReduce, an algebraic framework that reframes memory-efficient kernels like FlashAttention and Cut Cross-Entropy as "folds over commutative monoids". The authors present the "Local Gradient Theorem", which provides a sufficient condition for computing gradients efficiently by only using the final output and local inputs, thus avoiding the materialization of large intermediate tensors. The work is positioned as a "mainly theoretical" contribution that offers a principled, algebraic foundation for designing such layers, though it does not provide empirical performance benchmarks.

**Strengths:**

* The paper is strongly motivated by the need to move beyond ad-hoc engineering for memory-efficient kernels. It seeks to provide a "principled foundation" and "common mathematical structure" to unify recent advances like FlashAttention and Cut Cross-Entropy.

* The core direction of applying an algebraic framework to deep learning kernels is valuable. This "algebraic perspective" successfully unifies several disparate optimizations under the single concept of a monoidal fold, connecting modern ML kernels to classical parallel computing ideas like MapReduce.

**Weaknesses:**

* The paper's primary contribution appears to be an algebraic reframing of existing, well-known optimizations rather than the discovery of new ones. It explicitly states that FlashAttention can be seen as an instance of this general approach, and it also frames Cut Cross-Entropy and two-layer MLPs  within this structure. It is unclear if this new abstraction provides any new, practical, and efficient kernels beyond those that were already discovered through ad-hoc engineering.

* The paper is positioned as "mainly theoretical" and admits to focusing on "algebraic formulation... rather than empirical benchmarking". The authors' own proof-of-concept implementation is described as "neither parallel nor very efficient". This complete lack of empirical results makes it impossible to judge if the MonoidReduce abstraction offers any real-world advantages in terms of memory or wall-clock time over the highly optimized kernels it claims to generalize.

* The entire framework is built on commutative monoids. The paper explicitly admits this "excludes many common operations, such as (sequences of) matrix multiplications, which can be understood as associative, but not commutative, folds". This is a major restriction that severely limits the framework's applicability to a very small and specific set of neural network layers (primarily attention, cross-entropy, and a specific MLP formulation ).

* The core theoretical claim, the "Local Gradient Theorem" (Theorem 2.1), appears to be a simple consequence of the monoid's properties rather than a deep result. The condition d(x \odot y) / dx = D(x \odot y, x)  seems to be more of a definition of the specific class of monoids that allow this optimization, rather than a theorem that provides a new insight for a broad class of functions.

* The paper lacks a dedicated "Related Work" section that reviews the literature of algebraic construction/customization for ML models. Furthermore, Section 1.1, "WHAT ARE COMMUTATIVE MONOIDS?", is an undergrad-textbook-level definition that is highly inappropriate for the introduction of a top-tier conference paper and wastes valuable space that could have been used to better motivate the work or clarify its contribution.

* The paper uses notation that is atypical for a machine learning audience. For example, the type of the binary operation is written as $T \rightarrow T \rightarrow T$, which is a "curried" notation from functional programming, rather than the standard mathematical notation $T \times T \rightarrow T$. Additionally, the use of "backward/pullback Jacobian" notation for derivatives is non-standard and adds an unnecessary barrier to understanding (recall that "pullback" has a special meaning in algebraic functions/morphisms).

* The complexity analysis in Section 3 and the discussion clarify that the abstraction does not remove the difficulty of implementation. The authors state that a future framework would require "automatic tuning" of slicing and grouping parameters, which is the same complex, hardware-specific engineering challenge that the developers of "ad-hoc" kernels like FlashAttention must already solve. The paper provides a new language to describe this problem but offers no new solution for it.

**Questions:**

* The framework's reliance on commutative monoids is a significant limitation, as it excludes many common operations like standard matrix multiplication. How do you envision this framework being extended to the much broader and more common class of non-commutative (but still associative) operations, and what new challenges would arise for the Local Gradient Theorem in that setting?

* The paper is positioned as "mainly theoretical", and the provided proof-of-concept is admittedly "neither parallel nor very efficient". What is the concrete value of this framework beyond reframing existing, hand-optimized kernels like FlashAttention? Can you provide an example of a new, non-trivial, and practically useful kernel that was discovered because of the MonoidReduce abstraction?

* The complexity analysis and discussion make it clear that optimal performance still relies on hardware-specific tuning of slicing and grouping. Given that this is the primary challenge that "ad-hoc" engineering must solve, how does this algebraic abstraction actually help simplify or automate that core implementation problem?

---

> ### Author Response · Authors · 2025-11-24
>
> We thank Reviewer UHrH for the detailed and critical feedback. While we disagree on the assessment of our contribution's novelty and significance, we appreciate the opportunity to clarify our work.
>
> ## Regarding weaknesses:
>
> **Lack of empirical results**: We have addressed this in our general response. This is a theoretical paper, and our goal is to introduce the algebraic machinery, not a new SOTA kernel.
>
> **Limitation to commutative monoids**: See general response and below.
>
> **Local Gradient Theorem as "simple"**: We agree that the theorem is simple, but believe it should be judged by its utility rather than depth. While the proof is straightforward once the right abstractions are in place, the contribution lies in identifying these abstractions (monoidal folds) and formulating the precise condition that guarantees an efficient backward pass. This provides a clear, verifiable checklist for future kernel designers.
>
> **Related Work and Introduction**: Your point about the introductory material is well-taken. We will condense the definition of monoids and add a dedicated "Related Work" section that situates our framework within the broader context of algebraic methods in machine learning and systems in future versions.
>
> **Implementation difficulty**: Our framework does not eliminate the need for hardware-specific tuning, but it separates concerns. The implementer can focus on optimizing the monoid operator and the reduction schedule independently. The framework provides the guarantee that if the algebraic conditions are met, a memory-efficient implementation is possible.
>
> ## Regarding your questions:
>
> **Commutativity**: We discuss this in the general response. In short we note that the commutativity requirement is only put on the reduction: Both matrix multiplication and nonlinearities can be part of the mapping to the (commutative) monoid in question, as is done in the MLP-example. However, while we agree that associative scans are quite common in, for example, recurrent networks, we do not see associative folds (i.e. where intermediate results are only aggregated, and not fed into other downstreams layers) as all that common. For this reason we focus on commutative, associative, folds.
>
> **Concrete value beyond reframing**: We belive the value is threefold:
>
> - Unification: It provides a single lens to understand many different kernels.
> - Discovery: It provides a clear recipe for creating new memory-efficient layers.
> - Compositionality: Monoidal layers can be horizontally composed, preserving memory efficiency.
>
> While not discussed in the paper, a motivation for the work was in modelling conditional joint probabilities (edit: with cross-entropy) efficiently, i.e: `logprob(a, b | x) ~ <l(x, a), r(x, b)>` where the "vocabularies" of a and b are large. As far as we know, there is no existing memory efficient implementation of this, yet the fact that a memory efficient implementation exists is immediate from the abstractions introduced in this paper.
>
> **How abstraction helps implementation**: The abstraction helps by providing a formal contract. A systems engineer can be tasked with creating a highly optimized "MonoidReduce" implementation for a specific hardware target. A model designer, without needing to know the low-level details, can then create new memory-efficient layers simply by implementing the parts that the MonoidReduce abstraction expects, i.e. `proj_fold`, `binary_reduce`, `proj_fold_bwd`, e.t.c. This separation of concerns simplifies and accelerates development.

---

### Official Review · Reviewer_7UBs · 2025-10-30

**Soundness:** 3
**Presentation:** 3
**Contribution:** 2
**Rating:** 4
**Confidence:** 3

**Summary:**

The paper introduces MonoidReduce, an abstraction of several memory-efficient neural network kernels, such as FlashAttention, as folds over commutative monoids. The primary technical contribution is the Local Gradient Theorem, which provides a sufficient condition under which local computation is sufficient for computing a backward pass without storing large intermediate activations. The authors use this theorem to develop the abstract MonoidReduce protocol and apply it to several neural kernel components.

**Strengths:**

The framework is elegant, correct (to the best of my knowledge), and opens up a new directions for principled kernel design. The concept of modeling memory-efficient layers using commutative monoidal folds is highly creative. It unifies different optimized algorithms (like FlashAttention) under a single algebraic umbrella. The attempt to bridge abstract math with concrete hardware constraints (memory, IO, parallelism) is a necessary and valuable direction for deep learning research.

The Local Gradient Theorem (Theorem 2.1) provides a simple, easily verifiable condition that guarantees a memory-efficient backward pass. To my knowledge, this theorem is novel and correct. The abstraction for handling the backward pass is clean. It moves the complexity of gradient recomputation into an abstract and simplifies the overall training logic.

The explicit formulation of the Attention layer using the LogWSum monoid is a clear example of the framework's utility. It cleanly distills the salient properties of self-attention that makes it amenable to hardware sensitive implementations like FlashAttnetion.

**Weaknesses:**

The primary weaknesses lie in the development of the results and the demonstration of the framework's utility, especially since its core applications are already solved problems.

The main challenge of proposing a new abstraction over existing, highly optimized solutions (like FlashAttention) is justifying the abstraction itself. I recommend that the authors either:
* Broaden the Scope: Provide convincing arguments and theoretical analysis showing the potential for this methodology to unlock efficient implementations for other reductions that haven't received intense engineering effort. Examples include various sub-quadratic attention mechanisms, modern state-space models (SSMs), or other novel folding-based architectures.
* Provide Empirical Evidence: Provide empirical results that demonstrate the tangible benefits (e.g., speed and memory-savings) of constructions derived from MonoidReduce in contrast to highly-tuned alternatives, or at least show that the performance overheads are minimal across realistic hardware/tiling regimes.

The MonoidReduce algorithm (Listing 2) is presented as a Python code block in the appendix. This is the technical core of the paper and should be centered in the paper body without readers needing to parse large blocks of code.

While the framework is elegant and shows potential for systematizing memory-efficient parallel implementations of deep networks, the paper is currently presented as a theoretical proof-of-concept without strong empirical validation or detailed architectural implications. I lean toward reject but am open to reconsideration if the authors can address the substantial weaknesses regarding practical utility and scope.

These are minor points that should be addressed for clarity and presentation:
* Can the term "favorable tradeoff" (L418) be made more precise and concrete?
* A few statements on L356 and L392 are sentence fragments.

**Questions:**

The complexity analysis is a good starting point, but the results are difficult to evaluate without additional contextualization. There are several fundamental questions that could improve the theoretical completeness:
* In the case of attention, how do the quantities provided compare to Flash Attention?
* Are the resulting space and communication complexities for the backward pass optimal under the MonoidReduce assumptions?
* If the commutative assumption is removed (e.g., non-commutative folds), are similarly efficient, parallel implementations impossible?

---

> ### Author Response · Authors · 2025-11-24
>
> We thank Reviewer 7UBs for appreciating the elegance of our framework and the novelty of the Local Gradient Theorem. Your critique that we are addressing "already solved problems" is at the heart of the matter, and we appreciate you raising it.
>
> Our response to this core point is that providing a unifying abstraction for solved problems is a valuable contribution. It helps us understand why those solutions work and how to generalize them. That two-layer feed forward networks admit memory efficient versions is immediate from our work, and arguably immediate from the initial FlashAttention paper, yet efficient kernels are not abundant. To the best of our knowledge, [another paper](https://openreview.net/forum?id=rcsZNV9A5j) under review in this conference (no relation), which puts forward multi-head feed forward networks as a generalization of FlashAttention, is the first concrete memory efficient kernel of this kind.
>
> To address your questions:
>
> 1. **complexity vs FlashAttention**: The complexities of MonoidReduce applied to attention are asymptotically the same as FlashAttention2. Future versions will make this comparison explicit and provide a more in-depth discussion of low level hardware concerns.
> 2. **Optimality of the backward pass**: We believe there is space for significant constant improvements in the backward pass (e.g. grouping blocks of gradient computation and aggregation, thus reducing communication pressure), but that it is asymptotically optimal, we have, however, no proof of this.
> 3. **Commutativity**: We cover this in the general response. In short, commutativity ensures that we local values and final aggregate suffice for gradient computation. For non-commutative operations, the order of operations necessitates either a prefix or suffix as well, and thus requires more complex communication patterns. In future versions we will discuss this more in depth.

---

### Official Review · Reviewer_Cbgj · 2025-10-31

**Soundness:** 3
**Presentation:** 3
**Contribution:** 2
**Rating:** 4
**Confidence:** 2

**Summary:**

This paper introduces MonoidReduce a novel algebraic framework that unifies memory-efficient neural network layers like FlashAttention as folds over commutative monoids. The authors' core idea is that these specialized kernels are not ad-hoc solutions but share a common mathematical structure.

**Strengths:**

+ The paper is very well-written. The introduction to commutative monoids is clear and accessible. The detailed walkthrough of the attention mechanism as an instance of LogWSum is a good case study that makes the abstract theory concrete and easy to follow.

**Weaknesses:**

+ The paper is explicitly (and honestly) positioned as a theoretical contribution. There are no wall-clock time or memory usage benchmarks against the actual optimized kernels. The memory plots in Figures 3 & 4 are only compared to a naive PyTorch implementation, not a production-grade baseline, which makes it impossible to assess the practical viability or overhead of the proposed abstraction.

**Questions:**

+ Could the authors provide any empirical comparison, even if preliminary, of their proof-of-concept attention against the actual FlashAttention-2 kernel? Even if it's much slower, understanding the constant-factor overheads of the abstraction itself would be extremely valuable for assessing its practicality.

---

> ### Author Response · Authors · 2025-11-24
>
> We thank Reviewer Cbgj for the positive feedback on the clarity of our writing and presentation.
>
> You correctly point out that the main weakness is the lack of benchmarks against production-grade baselines. As we state in our general response, our focus was on introducing the theoretical framework. The memory plots against a naive implementation serve to validate the framework in the sense that it produces correct results using less memory. Your question about comparing our proof-of-concept to FlashAttention-2 is valid. A direct comparison would show our current implementation to be much slower, as it is written in pure Python/PyTorch without custom CUDA extensions.

---

### Official Review · Reviewer_BcXs · 2025-10-31

**Soundness:** 2
**Presentation:** 2
**Contribution:** 2
**Rating:** 4
**Confidence:** 4

**Summary:**

This paper introduces MonoidReduce, an algebraic framework that formalizes a general class of memory-efficient neural network layers as folds over commutative monoids. The key contribution is the Local Gradient Theorem, which supplies a sufficient condition for when the gradients of monoidal folds can be computed from final outputs and local recomputation, thereby enabling memory- and cache-efficient forward and backward passes. The framework unifies and extends the perspective behind recent engineering successes such as FlashAttention and Cut Cross-Entropy, offering a systematic foundation for discovering and implementing efficient neural network primitives. The paper is primarily theoretical, with proof-of-concept code and memory profiling that validate the approach on canonical neural net operations (attention, cross-entropy, MLPs), and provides an in-depth algebraic and complexity analysis.

**Strengths:**

1. The paper elegantly frames a range of recent memory-efficient neural network kernels under the umbrella of commutative monoid folds, connecting discrete algorithmic advances (e.g., FlashAttention, Cut Cross-Entropy) to a coherent algebraic foundation. This abstraction helps demystify why certain operator fusion or tiling schemes work, which encourages more systematic innovation beyond ad hoc engineering.

2. The introduction and formal statement/proof of the Local Gradient Theorem provide a clear and actionable condition for whether fold-based recomputation strategies are compatible with local gradient computation. This enables identifying previously undiscovered opportunities for memory-efficient implementation.

3. Memory profiling results in Figure 3 compared to Figure 4 substantiate the claim that folding over monoids can significantly reduce intermediate memory allocations, at least for the proof-of-concept implementations.

4. Section 3 delivers a nuanced, parameterized analysis of MonoidReduce’s computational, memory, and communication complexity, including both sequential and parallel regimes, and highlights system-level tradeoffs.

**Weaknesses:**

1. Although the theoretical complexity and memory profiles are suggestive, there is a conspicuous absence of actual wall-clock or throughput benchmarks with comparisons to highly optimized libraries (e.g., FlashAttention, Liger kernels, etc.). As acknowledged in Section 5 and throughout, the current code is proof-of-concept and “neither parallel nor very efficient.” This omission precludes a strong claim to practical impact and prevents full validation of the framework’s efficiency in large-scale, real-world settings.

2. The paper convincingly shows that many memory-efficient layers fit the monoid fold paradigm, but there is limited discussion and empirical exploration of cases that do not fit well, or of the limitations imposed when the commutativity or local gradient condition is violated. There are brief mentions of two-layer MLPs with certain dimensions and matrix multiplications, but a more systematic accounting (with empirical “what fails and why” cases) would greatly increase utility for practitioners.

3. The illustration is largely limited to attention-style layers and certain cross-entropy operations. While theoretical generality is discussed, it remains to be seen how broadly MonoidReduce extends to more exotic or domain-specific architectures commonly encountered in vision, speech, graph, or structured data ML.

4. While the algebraic and code presentation is precise, the writing is frequently dense (e.g., the full formalism for fold derivatives in Theorem 2.1 and the derivations in Section C), which may pose a barrier for practitioners not highly fluent in both category theory and modern autodiff conventions. There is little intuitive commentary or engineering “pseudocode” bridging the gap for those less comfortable with this notation.

5. Section 4 admits that proposed complexity tradeoffs may not always be favorable (e.g., in small-batch MLPs or when recomputation outweighs memory savings), but this is not systematically quantified. The claim that MonoidReduce is a key design abstraction may be strongest for a restricted set of layers (especially attention and related softmax-style reductions), rather than universally applicable.

6. While the importance of numerical stability (especially in log-space operations and parallel reductions) is briefly mentioned, practical guidelines on handling floating-point rounding, underflow/overflow, or automatic chunking are not fully fleshed out.

Potentially Missing Related Work:

[1] Jianfei Chen et al., ActNN: Reducing Training Memory Footprint via 2-Bit Activation Compressed Training.

[2] Tien Chu et al., Training DNNs in O(1) memory with MEM-DFA using Random Matrices.

[3] Bas Peters et al., Symmetric block-low-rank layers for fully reversible multilevel neural networks.

[4] Mohammad Rastegari et al., XNOR-Net: ImageNet Classification Using Binary Convolutional Neural Networks.

**Questions:**

1. Would the authors consider incorporating benchmarks of MonoidReduce (in e.g., a single-GPU and multi-GPU PyTorch/Triton implementation) compared to current state-of-the-art memory-efficient kernels such as FlashAttention, Liger, and ActNN? This would help clarify both achievable speedups and practical engineering tradeoffs.

2. Can the authors give concrete negative examples with actual benchmarks showing where MonoidReduce is not beneficial (for instance, when recomputation overhead outweighs memory savings), or where the commutative monoid structure breaks down?

3. What best practices or additional engineering, beyond the algebraic specification, do the authors recommend to handle floating-point instability, particularly in large/repeated reductions?

4. Have the authors explored or do they anticipate MonoidReduce being applicable to notable non-attention architectures (e.g., CNNs with grouped convolutions, spatial aggregations, or graph neural networks)?

5. Can more details or benchmarks be given as to how MonoidReduce interacts with prevailing accelerator architectures (e.g., shared memory management, multi-streaming, atomic updates)?

6. Will more detailed documentation, variable/parameter alignment, or a minimal working example be forthcoming to accelerate practitioner adoption?

**Details Of Ethics Concerns:**

N/A. No apparent use of sensitive data, unfair model biases, privacy, or research integrity issues.

---

> ### Author Response · Authors · 2025-11-24
>
> We thank Reviewer BcXs for the thorough and detailed summary of our work and the insightful feedback.
>
> ## Regarding questions:
>
> **Benchmark vs. SOTA**: we do not have benchmarks currently, as the proof-of-concept implementation is not accelerator-hardware aware. As discussed in the general response, we agree that this is a limitation.
>
> **Concrete negative examples**: In future versions we will add a more in-depth section on the tradeoffs of recomputation, as well as a more in depth discussion on the limits of the local gradient theorem and commutativity.
>
> **Floating point instability**: We envision two main general approaches to floating point instability: high-precision intermediates and deterministic reduction order. Both comes with tradeoffs, which will be discussed in future versions. In short, the most immediate approach is to have proj_fold output intermediate values in high precision, and perform the reduction in high precision. This should generally be acceptable, as the compute spent on reduction is dwarfed by the compute spent on mapping to the monoid. Deterministic reduction order is an orthogonal approach, where we lose the parallelization benefits of arbitrary reduction order, but gain stability due to the deterministic order of operations. Furthermore, our intuition is that the sensitivity to floating point instability varies quite significantly between monoids (e.g. max-pooling being less sensitive than logwsum, for example), and would suggest that users implement `binary_reduce`, `proj_fold`, et.c. in a manner that minimizes their susceptibility to such instabilities.
>
> **Applicability to non-attention architectures**: While MLPs are discussed in the paper, there is not much discussion on other or novel types of layers. We believe the framework can be applied to GNNs and grouped convolution and convolution. In the case of convolutions, special care must, however, be taken w.r.t. slicing.
>
> **Accelerator architecture**: In future versions we will elaborate on how the tiling and streaming inherent to MonoidReduce map naturally to GPU shared memory and multi-streaming capabilities.
>
> **Documentation**: The supplementary material contains a minimal working example, as well as implementations of the layers discussed in the paper. The implementation is however, as stated in the paper, neither parallel nor very efficient.

---

### Author Response · Authors · 2025-11-24
**General Response**

We thank all the reviewers for their time and for providing detailed, constructive feedback on our work. We are encouraged that reviewers found the framework and formulation "elegant" (7UBs, BcXs), and a "valuable direction for deep learning research", (7UBs, UHrH).

# Theoretical or Practical?
The central critique, shared by all reviewers, is the lack of empirical, wall-clock benchmarks against highly-optimized kernels like FlashAttention. We acknowledge this limitation and want to clarify our contribution. This paper is intended as a theoretical work, with a proof of concept implementation to validate the theory. Our primary goal is not to introduce new state-of-the-art kernels, but rather to introduce a unifying algebraic framework, supported by the Local Gradient Theorem, that provides a principled foundation for designing such kernels. We believe that moving from the current paradigm of making existing layers (or sequences of layers) memory efficient *after the fact* to a paradigm where models/layers are intentionally designed for memory efficiency is an important direction for the field, and one which our paper offers significant theoretical tooling for.

While our proof-of-concept code is not optimized for performance, it serves to validate the correctness of the framework and its memory-saving potential, as well as demonstrate the separation of concerns between the framework and framework user. We agree that demonstrating the practical utility of this abstraction is a critical next step.

# Commutativity?
A critique raised by both BcXs and UHrH is that the commutativity requirement significantly limits the applicability of the framework. We agree that it is a limitation, but would like to offer some comments.

The first comment is that the commutativity requirement is imposed only on the reduction. This leaves significant freedom in the mapping to the monoid. As demonstrated by the MLP example the mapping can involve both matrix multiplications and nonlinearities.

Second, there exists a Local Gradient-esque theorem for associative folds, but it requires either an accumulated prefix or suffix: if $\frac{d ~ x\odot y }{d ~ y}=D_r(x\odot y, y)$, and $\frac{d ~ x\odot y }{d ~ x}=D_l(x\odot y, x)$, then for $p_i = x_1 \odot x_2 \odot \ldots \odot x_i$ we have that for all $j\leq i$,  $\frac{d ~ p_i}{d ~ x_j} = D(p_i, p_j, x_j)$, where $D(p_i, p_j, x_j) = D_r(p_j, x_j) \cdot D_l(p_i, p_j)$. The backwards pass is therefore not as immediately parallelizable as in the commutative case, where the final fold and local values suffice. The forward pass is also less immediately parallelizable as the order of operations must be taken into account during the reduction.

Finally, while associative **scans** are common in, for example, recurrent models, associative **folds** are mostly found in their most general form: function/layer composition. Function composition does not in general admit compact representation, nor does it in general satisfy the local gradient criteria.

---

### Meta-Review · Area_Chair_xTMq · 2026-01-04

**Summary:**

Reviewer BcXs: The paper elegantly frames a range of recent memory-efficient neural network kernels under the umbrella of commutative monoid folds, connecting discrete algorithmic advances to a coherent algebraic foundation. The introduction and formal statement/proof of the Local Gradient Theorem provide a clear and actionable condition. Memory profiling results in Figure 3 compared to Figure 4 substantiate the claim that folding over monoids can significantly reduce intermediate memory allocations. However, the reviewer still has some concerns on the weaknesses about conspicuous absence of actual wall-clock or throughput benchmarks with comparisons to highly optimized libraries, limited discussion and empirical exploration of cases that do not fit well, or of the limitations imposed when the commutativity or local gradient condition is violated.

Reviewer Cbgj: The paper is very well-written. The introduction to commutative monoids is clear and accessible. The detailed walkthrough of the attention mechanism as an instance of LogWSum is a good case study that makes the abstract theory concrete and easy to follow.  However, the reviewer still has some concerns on the weaknesses about lack of wall-clock time or memory usage benchmarks against the actual optimized kernels, not a production-grade baseline.

 Reviewer 7UBs: The concept of modeling memory-efficient layers using commutative monoidal folds is creative. The Local Gradient Theorem (Theorem 2.1) provides a simple, easily verifiable condition that guarantees a memory-efficient backward pass. However, the reviewer still has some concerns on the weaknesses about  the development of the results and the demonstration of the framework's utility, lack of empirical evidence.

Reviewer UHrH: The paper is strongly motivated by the need to move beyond ad-hoc engineering for memory-efficient kernels.  The core direction of applying an algebraic framework to deep learning kernels is valuable. However, the reviewer still has some concerns on the weaknesses about lack of novelty and a dedicated "Related Work" section,  lack of theoretical depth, unclear descriptions.

**Reviewer Concerns:**

After carefully evaluating the rebuttals, I think the reviews from the Reviewer  UHrH  were partially addressed from the response.
For the remaining reviewer concerns, they are all not fully addressed.

**Reviewer Scores:**

For reviewer UHrH , if the reviewer had been able to participate the discussion, I think the reviewer may slightly increase the rating.

For all the remaining reviews, if the reviewer had been able to participate the discussion, I think the reviewer may keep the original rating unchanged or decrease the rating.

Based on the rebuttal, lots of concerns were not successfully addressed.

---

### Decision · Program_Chairs · 2026-01-26

Reject